# The Dynamics of Changes in the Concentration of IgG against the S1 Subunit in Polish Healthcare Workers in the Period from 1 to 12 Months after Injection, Including Four COVID-19 Vaccines

**DOI:** 10.3390/vaccines10040506

**Published:** 2022-03-24

**Authors:** Monika Skorupa, Joanna Szczepanek, Agnieszka Goroncy, Joanna Jarkiewicz-Tretyn, Barbara Ptaszyńska, Paweł Rajewski, Wojciech Koper, Krzysztof Pałgan, Andrzej Tretyn

**Affiliations:** 1Faculty of Biological and Veterinary Sciences, Nicolaus Copernicus University, 87-100 Torun, Poland; 2Centre for Modern Interdisciplinary Technologies, Nicolaus Copernicus University, 87-100 Torun, Poland; szczepanekj@umk.pl (J.S.); prat@umk.pl (A.T.); 3Faculty of Mathematics and Computer Science, Nicolaus Copernicus University, 87-100 Torun, Poland; gemini@mat.umk.pl; 4Non-Public Health Care Centre, Cancer Genetics Laboratory, 87-100 Torun, Poland; prezes@genetykatorun.pl; 5Marshal’s Office of the Kuyavian-Pomeranian Voivodeship, 87-100 Torun, Poland; b.ptaszynska@kujawsko-pomorskie.pl; 6Department of Internal and Infectious Diseases, Provincial Infectious Disease Hospital, 85-067 Bydgoszcz, Poland; rajson@wp.pl; 7The Voivodeship Sanitary-Epidemiological Station in Bydgoszcz, 85-031 Bydgoszcz, Poland; wkoper@pwisbydgoszcz.pl; 8Department of Allergology, Clinical Immunology and Internal Diseases, Collegium Medicum, Nicolaus Copernicus University, 85-067 Bydgoszcz, Poland; palgank@cm.umk.pl

**Keywords:** SARS-CoV-2, COVID-19, vaccination, antibodies, BNT162b2, mRNA-1273, ChAdOx1 nCoV-2019, Ad26.COV2.S

## Abstract

Background: The presented research made it possible to obtain the characteristics of changes in anti-SARS-CoV-2 IgG within one year of vaccination in healthcare workers. Materials and Methods: The research group consisted of 18,610 participants represented by medical and administration staff. IgG antibody concentrations were determined by ELISA. Results: At 5–8 months after full vaccination, the levels of anti-SARS-CoV-2 IgG with equal vaccines were similar. The exception was JNJ-78436735, for which IgG levels were significantly lower. In the 9th month after vaccination, an increase in the anti-SARS-CoV-2 IgG level, suggesting asymptomatic infection, was observed in a large group of participants. Significantly higher levels of anti-SARS-CoV-2 IgG antibodies were observed after the booster dose compared to the second dose. The increase in antibodies was observed already around the 5th day after the injection of the booster dose, and was maximized at approximately the 14th day. Conclusion: The cut-off date for protection against the disease seems to be the period 8–9 months from the vaccination for mRNA vaccines and 5–6 months for vector vaccines. The introduction of a booster dose was the right decision, which could have a real impact on restricting the further transmission of the virus.

## 1. Introduction

The severe acute respiratory syndrome, coronavirus 2 (SARS-CoV-2), the virus responsible for the coronavirus disease 2019 (COVID-19), has infected over 440 million people worldwide, and caused the deaths of over 5.97 million. In Poland, the first case of COVID-19 was announced on 4 March 2020. Since then, the total number of cases has reached over 5.69 million, and approximately 112,000 people have died [1].

Vaccination against COVID-19 in Poland first began on 27 December 2020 with healthcare workers under the so-called National Vaccination Program, and then gradually continued with the further population, depending on age. From June 2021, children in the age range of 12–17 years began to be vaccinated, and from December 2021, those in the range of 5–11 years. After nearly a year, more than 53 million doses of anti-COVID-19 vaccines were administered. It is estimated that over 22 million people are fully vaccinated in Poland, which is approximately 58.5% of the population. Importantly, over 30% of them took the third dose of the vaccine, i.e., the booster dose [2].

The introduction of preventive vaccination now seems to be the best way to fight the SARS-CoV-2 pandemic, reducing the risk of severe COVID-19 and the number of hospitalizations and deaths due to the disease, and bringing the pandemic closer to its end [2,3]. So far, four types of anti-SARS-CoV-2 vaccines have been administered in Poland. The BNT162b2 Comirnaty (Pfizer/BioNTech) and mRNA-1273 Spikevax (Moderna) vaccines involve mRNA technology. The nucleoside-modified messenger RNA contained in the vaccine is encapsulated in lipid nanoparticles, allowing non-replicating RNA to pass into the host cells to allow for transient expression of the SARS-CoV-2 virus S antigen. BNT162b2 has been available in Poland since 27 December 2020. It can be taken by people over 12 years of age, with an interval of 21 days between doses. In mid-September 2021, the third dose of the vaccine was authorized, which was initially intended for people aged 50+ and medical service workers. The effectiveness of this vaccine in preventing infection is 94%, measured over seven days to six months after the second dose. The vaccine was also 100% effective in preventing severe disease and death [4]. Vaccinations with mRNA-1273 began in Poland on 20 January 2021. They are intended for people over 18 years of age, and the second dose should be taken after 28 days. The vaccine is 95% effective in preventing disease from 14 days after the second dose, and is also 100% effective against severe disease [5]. The next two preparations of ChAdOx1 nCoV-2019, (AstraZeneca) and Ad26.COV2.S (Janssen/Johnson & Johnson) are vector vaccines. ChAdOx1 nCoV-2019 is a monovalent vaccine consisting of a single, recombinant, replication-deficient chimpanzee adenoviral vector, encoding the SARS-CoV-2 virus S glycoprotein. In Poland, this preparation can be used to vaccinate people over 18 years of age as of 12 February 2021, and the interval between doses is 28 to 84 days. The effectiveness of ChAdOx1 nCoV-2019 in preventing COVID-19 is already much lower than that of mRNA vaccines, and amounts to 70–79%. In principle, it protects against severe disease in 100% of cases [6]. The Johnson & Johnson vaccine is also a monovalent vaccine consisting of a recombinant, replication-free human adenoviral vector type 26, encoding the full-length SARS-CoV-2 S glycoprotein. The preparation is also intended for adults, and the single-dose was introduced for use on April 14, 2021. The effectiveness of the Ad26.COV2.S is the lowest among all vaccines. In the prevention of infection, it is at the level of 66–72%; in the protection against the severe course of COVID-19, it is at the level of 85% [7]. According to the literature, 6 months after vaccination, the effectiveness of preventing infection significantly decreases with all vaccines. For BNT162b2, it is 45%; for mRNA-1273, it is 58%; for Ad26.COV2.S, it is 13%. Protection against severe COVID-19 and death after this time is also lower, although these values remain relatively high; for BNT162b2 it is 70%; for mRNA-1273, it is 76%; and for Ad26.COV2.S, it is 52% [8].

In Poland, the BNT162b2 vaccine has dominated from the beginning. It accounts for 69.8% of all vaccines used in Poland, followed by ChAdOx1 nCoV-2019 at 16.75%, mRNA-1273 at 10.40%, and Ad26.COV2.S at 2.98% [9]. According to data from the Ministry of Health, as of December 9, 2021, 83.1% of healthcare workers are fully vaccinated, of which doctors make up 92%, dentists make up 87%, laboratory diagnosticians make up 85%, midwives make up 82%, nurses make up 81%, and physiotherapists make up 71% [10].

With the introduction of COVID-19 vaccination, the number of people hospitalized due to COVID-19 has decreased, which currently accounts for approximately 20–30% among fully vaccinated people. The number of death cases has also decreased. According to data from the Ministry of Health, during the period from 2 January to 13 December 2021, for those fully vaccinated with Pfizer/BioNTech with comorbidities (and without comorbidities), 0.02% (and 0.005%) died from COVID-19, while the rates were 0.01% (0.003%) for AstraZeneca, 0.01% (0.002%) for Moderna, and 0.01% (0.003%) for Johnson & Johnson [9].

Current serological tests allow for the determination of SARS-CoV-2-specific antibodies directed against the S protein and the nucleocapsid (N) protein [10,11]. In people who have already suffered from COVID-19, IgG antibodies are directed against various proteins of the virus. In vaccinated individuals who were seronegative prior to vaccination, IgG antibodies are only directed against the protein S. The S protein is homotrimeric, and each monomer consists of two subunits: S1 and S2. The S1 subunit consists of two domains: N-terminal and C-terminal. The C-terminal domain is the receptor-binding domain (RBD) that recognizes and binds to a receptor on the cell surface—the angiotensin-converting enzyme receptor 2 (ACE2). The N-terminal domain is involved in the initial binding of the virus to cells. The S2 subunit and the fusion peptides are responsible for the fusion of the viral membrane and the host cell. Their release is made possible only by the activity of cellular enzymes [10,11,12]. The S1 subunit contains an immunologically relevant receptor binding domain, which is a key target of neutralizing antibodies. The S1 subunit also has the lowest homology with analogous regions of other human pathogenic coronaviruses, suggesting that it may show less cross-reactivity [12].

On the basis of numerous scientific studies on the short-term vaccine response, the time from full vaccination at which the highest immunological parameters, and thus the highest protection against infection, are achieved was characterized. However, research is still ongoing to determine the duration of this protective period. Despite the vaccine’s high efficacy, there is still much more to learn about how individual groups respond to the vaccine. Continuous monitoring post-vaccination is essential to be able to determine the duration of the protection period and the timing of the booster dose. A very important question is also whether a permanent protective effect can be obtained. That is why it is so important to study the long-term response. The research group of the presented work is represented by employees of medical entities. Most of this group was vaccinated in the so-called group “0”; furthermore, they are regularly screened for SARS-CoV-2 infection, and were also the first to be able to take a booster dose. Therefore, this group is an excellent research source for characterizing the dynamics and stability of ant-SARS-CoV-2 IgG levels in response to vaccination. This paper presents a comparison of the dynamics and persistence of anti-SARS-CoV-2 IgG levels in response to vaccination with the four vaccines available in Poland (i.e., BNT162b2, mRNA-1273, ChAdOx1 nCoV-2019, and Ad26.COV2.S) 1–12 months after full vaccination and 1–3 months after booster dose administration.

## 2. Materials and Methods

### 2.1. Participants Group Characteristics

Analysis of a selected marker of a post-vaccination response was carried out as part of the European Social Fund project: “Minimizing Impact of COVID-19 Infection on Medical Staff Through Protecting Prophylaxis”. The project was created in order to take preventive and protective measures that would reduce the negative effects of COVID-19. The implementation of the program began on 1 March 2020 and its continuation is planned until 30 June 2022. Twenty-nine medical entities from the entire voivodship are cooperating in the partnership between the Kuyavian-Pomeranian Voivodeship Self-Governments. The current study takes into account the data obtained during diagnostic tests performed by the end of 2021. The recipients of the project are employees of the abovementioned facilities as well as medical entities providing care under the National Health Fund in the voivodeship. The effect of the activities carried out is that, from June to December 2021, 18,838 serological tests were performed on employees of medical entities. An additional effect, extremely important to scientific research, is the possibility of analyzing the obtained results, which constitute the basis of this study.

Out of the total number of 18,838 tests performed, 228 results were excluded from further analysis. The reasons for exclusions were incomplete data, most often including missing vaccination dates or SARS-CoV-2 infection. The research group consisted of 18,610 people represented by employees of medical entities, including doctors, nurses, midwives, paramedics, and laboratory diagnosticians, as well as administration and support staff. There was a large disproportion in the number of participants in terms of gender, as 81.3% were women and 18.7% men (Table 1). The most numerous group, 33.3%, was represented by people aged 46–55, and the smallest group was 3.9%, represented by people over 66 years of age. Study participants were injected with 4 different available COVID-19 vaccines at different times. However, the most numerous subgroup, constituting almost 83% of the respondents, was the one represented by individuals vaccinated with the BNT162b2. The second largest subgroup studied were people vaccinated with the ChAdOx1 nCoV-2019 (5.2% of the participants; Table 1), followed by 3191 people who took the third dose of the Pfizer/BioNTech vaccine between mid-September and mid-December.

### 2.2. IgG Anti-SARS-CoV-2 Assays

The anti-SARS-CoV-2 IgG level was determined by enzyme-linked immunosorbent assay (ELISA) using the automatic analyzer EUROIMMUN Analyzer I-2P and the Anti-SARS-CoV-2 QuantiVac ELISA kit (IgG) EUROIMMUN (Lübeck, Germany) according to the manufacturer’s instructions and a detailed description of the methodology from our previous research (Tretyn et al. 2021 [13]). The assays were performed on 96-well microplates coated with the SARS-CoV-2 S1 domain (including RBD) recombinantly expressed in the human HEK293 cell line (ATCC).

### 2.3. Statistical Analysis

Statistical procedures and calculations were performed with the use of the R environment (v4.1.2) and IBM SPSS Statistics (v27), and all statistical tests were considered to be statistically significant if their respective *p*-values were less than 0.05.

First, the unvaccinated group of subjects who did not report the prior COVID-19 infection was considered. In order to estimate the percentage of those who had recovered from COVID-19 without knowing it (based on significantly high IqG concentrations at a given time point), a Student’s *t*-test for independent samples was conducted, and a 95% confidence interval was estimated. Next, the group of fully vaccinated, whose antibody level was measured at least 6 months after the second dose, declaring no prior COVID-19 infection, was analyzed. The percentage of the respective group of COVID-19 recoverers was also tested with Student’s *t*-test for independent samples and 95% confidence interval was estimated.

Further, the group of subjects vaccinated with three doses of vaccine was included, and vaccinated subjects were divided into three groups, depending on the number of doses, the antibody level, and the declaration of prior COVID-19 infection. The comparison between the antibody levels in these three groups was performed a the one-way analysis of variance (referred to in this paper as ANOVA) and post hoc Games–Howell tests.

Next, for subjects with the BNT162b2 vaccination, we compared the means of the antibody titer between two groups: vaccinated with only 2 doses no more than 3 months ago and the analogous group vaccinated with 3 doses, depending on the COVID-19 infection status, using the two-way ANOVA (group factor: 2 levels; COVID-19 factor: 2 levels) and multiple comparisons for simple effects, which were supported by the bootstrap method based on 2000 replicates.

However, the main part of the analysis was to compare the mean antibody levels for each of the vaccine type depending on time elapsed after the full vaccination. For this purpose, the two-way ANOVAs (time factor: 3–4 levels; COVID-19 factor: 2 levels) comparing the means of the antibody level were conducted for each type of the vaccine. In the case of the time main effect being significant, further post hoc Hochberg’s GT2 tests were conducted. In order to break down the interaction effects (when justified), multiple comparisons for simple effects analyses were also performed. All these procedures were supported by the bootstrap method based on 2000 replicates, since for every vaccine type the distributions of the antibody levels were right-skewed and respective groups of observations were unbalanced. Before the ANOVAs were undertaken, the extremely large values (outliers) (i.e., those exceeding the upper quantile by three interquartile ranges in each group determined by the time after vaccination and prior COVID-19 infection) were removed from the analysis (see Table 2 and Table 3 for details).

## 3. Results

### 3.1. Characteristics of Unvaccinated Participants

Out of 18,610 participants, 1565 employees (8.41%) were not vaccinated. Of these, 621 recovered from SARS-CoV-2 (based on a positive qPCR test). Fifteen people were suspected of an infection, and five had reinfection. There are 941 subjects who were not vaccinated, declaring that they did not have the COVID-19 infection. For 385 of these respondents, the results of the antibody level were higher than 35.2 BAU/mL (cut-off value of the positive result). The results of a *t*-test (*t*(940) = −0.054, *p*-value = 0.957) showed that we could not reject the null hypothesis that 41% of healthy unvaccinated subjects had recovered from COVID-19. The 95% confidence intervals for the percentage of SARS-recoverers among the unvaccinated were 37.77% and 44.06%. A summary of the antibody levels is presented in Table 4.

### 3.2. Comparison of Antibody Titers among Vaccines

In the first four months after full vaccination, a high dynamic of change in the level of anti-SARS-CoV-2 antibodies was observed, especially in the case of mRNA vaccines. These levels were higher in those who were seropositive before vaccination compared to those who were not infected. The mean IgG titers during these months for mRNA vaccines in seronegative subjects were 3004.96, 2067.00, 1342.04, and 975.90 BAU/mL, respectively. In the seropositive subjects, they were 4358.83, 3022.72, 1701.88, and 1590.61 BAU/mL, respectively, for the BNT162b2 vaccine, and 4405.68, 3537.75, 1643.6, and 1460.96 BAU/mL, respectively, for the mRNA-1273 vaccine in subjects without SARS-CoV-2 infection. In pre-vaccine infected persons, they were 8079.23, 4398.72, 2551.25, and 1230.23 BAU/mL (Table 3). The mean IgG values for vector vaccines in respondents without COVD-19 were 706.5, 526.37, 287.26, and 271.72 BAU/mL for those receiving the ChAdOx1 nCoV-2019 vaccine, and 1022.15, 1208.41, 720.98, and 745.59 BAU/mL for those receiving Ad26.COV2.S. For convalescents, on the other hand, the mean values were 1065.56, 1136.3, 790.46, and 586.75 BAU/mL and 2085.63, 2300.33, 959.45, and 1120.15 BAU/mL. From 5 to 8 months after full anti-SARS-CoV-2 IgG vaccination, the levels approached each other, and were more stable with the tested vaccines, with the exception of Ad26.COV2.S, where the values were significantly lower. The mean antibody titers after vaccination with mRNA vaccines in respondents without COVID-19 were 627.01, 543.20, 483.31, and 328.62 BAU/mL for BNT162b2 and 1241.54, 1038.70, 1120.12, and 1052.66 BAU/mL for mRNA-1273, and in convalescents in such a system, they were 1180.34, 871.58, 811.35, and 707.54 BAU/mL and 1893.14, 1359.86, 760.04, and 691.53 BAU/mL. For vector vaccines, the mean antibody levels in seronegative subjects were 328.80, 1002.52, 1384.63, and 348.94 BAU/mL and 634.46, 1080.98, 1535.15, and 1671.23 BAU/mL for ChAdOx1 nCoV-2019 and Ad26.COV2.S, respectively; for seropositive respondents on the same system, they were 893.36, 855.43, 1402.78, and 422.00 BAU/mL and 804.31 (5th month), 1345.10 (6th month), and 790.33 BAU/mL (7th month). It should be added that the presented mean IgG over this time period for the vector vaccines does not reflect the discussed stabilization. This time coincides with the introduction of a booster dose of the vaccine and the ongoing fourth wave of the COVID-19 pandemic. Many observations were characterized by high values of antibodies, exceeding the mean values, caused by the administration of the next dose of the vaccine or infection with SARS-CoV-2. For mRNA vaccines, the reporting period is generally from 9 months after full vaccination. A significant increase in the level of anti-SARS-CoV-2 IgG was observed. The mean antibody titers for BNT162b2 from month 9 to month 12 in the seronegative were 753.22, 1952.48, 2549.28, and 2391.13 BAU/mL, and those in the seropositive group, from the 9th to 11th months, were 1109.24, 2155.8, and 2284.43 BAU/mL, respectively. In contrast, in the case of the mRNA-1273 vaccine, in the 9–10-month period after vaccination, the mean antibody values were 2294.25 and 242.95 BAU/mL in seronegative respondents, and 908.6 and 1374.9 BAU/mL in convalescents (Table 5).

### 3.3. Comparison of the Differential Concentration of Antibodies in Fully Vaccinated Subjects 6 Months after Injection

Since October, we noticed large variations in antibody levels among people challenged with two doses of BNT162b2. Based on the survey data and IgG concentrations, we decided to distinguish and compare three groups (Figure 1):Group 1—Vaccinated (two doses) more than 6 months before the antibody measurement, low antibodies level (<2500), regardless of the declaration of COVID-19 prior infection (*N =* 9181). In this group, we observed a systematic decrease in IgG concentration, which was a natural consequence of the lack of subsequent immunization;Group 2—Vaccinated (two doses) more than 6 months before the antibody measurement, high antibodies level (>2500), and no declaration of COVID-19 prior to infections (*N =* 190). In this group, we observed a significant increase in IgG concentration, probably a consequence of an unconscious (asymptomatic) SARS-CoV-2 infection;Group 3—Vaccinated (three doses), regardless of the declaration of COVID-19 prior to infection and antibodies level (*N =* 3191). Similar to the second group, we noted a significant increase in IgG levels as a result of the third injection.

Among fully (two-dose) vaccinated subjects (*N =* 13,648), there were 6620 whose antibody levels were measured at least 6 months after the last dose of the vaccine and did not declare a COVID-19 prior to infection (Table 6). In this group, we extracted 190 subjects with antibody levels exceeding 2500. The *t*-test result showed that in this population of those vaccinated early and without COVID-19 infection symptoms, 2.9% (95% CI: (2.468%, 3.272%)) were actually SARS-CoV-2-infected (*t(*6619) = −0.146 *p*-value = 0.884).

### 3.4. Changes in Antibody Levels over 12 Months after Vaccination with BNT162b

There were 15,237 subjects vaccinated with two or three doses of Pfizer/BioNTech. First, 982 outliers exceeding the upper quantile by three interquartile ranges in groups defined by the time elapsed after full vaccination (≤3 months, 3–6 months, 6–9 months, and 9+ months) and COVID-19 prior infection (no, yes) were rejected (see Table 7). Finally, the data counted 14,255 observations.

There was a significant main effect of time elapsed after full vaccination (*F*(3, 14247) = 1356.531, *p*-value < 0.001, η^2^ = 0.222). The changes in the level of antibodies were the most dynamic during the first 3 months. In the first month after taking the second dose of BNT162b, the level of antibodies increased intensively, followed by a systematic decrease in the concentration of anti-SARS-CoV-2 IgG (Table 3). The dynamics of changes were the highest in the first months after full vaccination, followed by a relative stabilization of the dynamics of changes, from approximately 5 to 8 months. After this period, we observed an increase in the level of antibodies as a consequence of subsequent SARS-CoV-2 infections and the introduction of vaccination with the third dose (described below). The bootstrap post hoc tests revealed that the antibody level ≤3 months after vaccination was significantly higher than at 3–6 months and 6–9 months (both *p*-values < 0.001), and significantly lower than at 9+ months (*p*-value < 0.001). Moreover, the antibody levels at 3–6 and 6–9 months were also significantly lower than at 9+ months (both *p*-values < 0.001). There were no significant differences in the antibody levels between 3–6 months and 6–9 months (*p*-value = 0.937). This can be seen in Figure 2A.

Prior SARS-CoV-2 infection was one of the main factors influencing antibody levels among participants regardless, of the time of the vaccination. There was a significant main effect of COVID-19 (*F*(1, 14247) = 112,389, *p*-value < 0.001, η^2^ = 0.008). Subjects with prior COVID-19 infection had significantly higher level of antibodies than those without (see Figure 2B). What is most interesting is that there was a significant interaction between the time elapsed after vaccination and prior COVID-19 infection (*F*(1, 14247) = 49.651, *p*-value < 0.001, η^2^ = 0.010). In order to break down the interaction effects, a simple effects analysis was conducted. The results of the bootstrap method for pairwise comparisons indicate that the antibody level in time units was affected differently depending on prior COVID-19 infection. Specifically, significant differences were revealed for every group (*p*-values < 0.001), with the following exceptions. Subjects with no prior COVID-19 infection did not differ in antibody levels between 3–6 months and 6–9 months (*p*-value = 0.455) after vaccination. Those who recovered from COVID-19 also had similar antibody titers when at 3–6 months and 6–9 months (*p*-value = 0.093), and additionally when comparing ≤3 months versus 9+ months (*p*-value = 0.104) after vaccination. Finally, there were significant differences for all time units after full vaccination between subjects with and without prior COVID-19 infection (*p*-values < 0.006): for each time interval, the antibody titer was significantly lower when no COVID-19 infection was reported, except when we considered 9+ months, at which point the antibody level was significantly lower in the prior COVID-19 infection group. These conclusions are illustrated in Figure 2B and Table 5.

One can wonder why in the last group, 9+ months after full vaccination, a significant increase in antibody titer is noticeable compared to the previous group. This is probably caused by other factors, such as the third dose of vaccine or the latent COVID-19 infection, as can be seen in Figure 3.

There are also 3191 three-dose vaccinated subjects in the data set, regardless of prior COVID-19 infection and antibody level (see Table 6 for details). In order to compare the mean levels of antibodies in the three groups, one-way ANOVA with the group factor was conducted. Its results allow us to state that there were significant differences between mean values of antibody levels in the groups (*F*(2, 12,559) = 7178, *p*-value < 0.001, η^2^ = 0.53). Further post hoc analysis showed that there were significant differences between every pair of groups (*p*-values < 0.001). The lowest antibody level was noticed in the first group, while this was significantly higher for the third group, and highest in the second group (Table 8 and Figure 3).

### 3.5. Comparison of the Two Doses under 3 Months Vaccinated Group and the Three Doses under 3 Months Vaccinated Group

We compared the mean antibody titer in the groups of BNT162b2 vaccinated with only two doses (954 observations) no fewer than 3 months prior with the analogous group vaccinated with three doses (3132 observations), depending on the COVID-19 infection report. The mean antibody titers in the first group were significantly lower than in the second group (*F*(1, 4082) = 183.685, *p*-value < 0.001, η^2^ = 0.043) as shown on Figure 4, and the mean for those without COVID-19 infection was significantly lower than for those reporting COVID-19 (*F*(1, 4082) = 12.013, *p*-value < 0.001, η^2^ = 0.003) (see Figure 4). The interaction between the groups and COVID-19 was also significant (*F*(1, 4082) = 19.206, *p*-value < 0.001, η^2^ = 0.005). Both groups differed in mean antibody titers: means in the first group were significantly lower than in the second group for COVID-19 recoverers, as well as for those without infection (*p*-values < 0.001). Moreover, in the first group, the mean titers for subjects without COVID-19 infection were significantly lower than for those who reported COVID-19 positivity (*p*-value < 0.001); however, in the second group, the effect of COVID-19 was not significant (*p*-value = 0.425). These implications are illustrated in Figure 4.

### 3.6. Changes in Antibody Levels over 10 Months after Vaccination with mRNA-1273

There were 274 subjects with two or three doses of the mRNA-1273 vaccinate. For Moderna, we observed the highest antibody levels in each of the compared months (Table 6). The dynamics of the change profiles in the subsequently analyzed periods were similar to those of Pfizer/BioNTech.

First, 11 outliers exceeding the upper quantile by three interquartile ranges in the groups defined by time elapsed after full vaccination (≤3 months, 3–6 months, and 6–9 months) and prior COVID-19 infection (no, yes) were rejected (Table 7). Since there was only one observation in the group 9+ months after full vaccination, this was also removed from analysis. Finally, the data counted 262 observations. There was a significant main effect of time elapsed after full vaccination (*F*(2, 256) = 3.402, *p*-value < 0.035, η^2^ = 0.026). The antibody level at ≤3 months after vaccination was significantly higher than at 3–6 months and 6–9 months (both *p*-values < 0.001). There was no significant difference in the antibody levels between 3–6 months and 6–9 months (*p*-value = 0.995). This can be seen in Figure 5 and Table 5. The main effect of COVID-19 was not significant (*F*(1, 256) = 0.274, *p*-value = 0.601), and neither was the interaction between the time elapsed after vaccination and the prior COVID-19 infection (*F*(2, 256) = 0.174, *p*-value = 0.840).

### 3.7. Changes in Antibody Levels over 8 Months after Vaccination with ChAdOx1 nCoV-2019

There were 959 participants vaccinated with AstraZeneca. In June, a decision was made in Poland to suspend vaccinations with AstraZeneca, and only the second dose was administered in people who started this vaccination course. Participants in this group took Pfizer/BioNTech as the third dose. First, 57 outliers exceeding the upper quantile by three interquartile ranges in groups defined by the time elapsed after full vaccination (≤2 months, 2–4 months, 4+ months) and prior COVID-19 infection (no, yes) were rejected (see Table 9). Finally, the data counted 902 observations.

There was a significant main effect of time elapsed after full vaccination (*F*(2, 896) = 23.847, *p*-value < 0.001, η^2^ = 0.051). The post hoc tests were significant for all pairs of time intervals, and revealed that the antibody level ≤ 2 months after vaccination was significantly higher than at 2–4 months and 4+ months (both *p*-values < 0.001), while the antibody levels at 2–4 months were also significantly lower compared to 4+ months after full vaccination (*p*-value = 0.036). This result is presented in Figure 6A. There was a significant main effect of COVID-19 (*F*(1, 896) = 90,353, *p*-value < 0.001, η^2^ = 0.092). Subjects with prior COVID-19 infection had significantly higher levels of antibodies than those without (see Figure 6B).

Similar to mRNA vaccines, we obtained a significant interaction between the time elapsed after ChAdOx1 nCoV-2019 vaccination and prior COVID-19 infection (*F*(2, 896) = 4.048, *p*-value = 0.018, η^2^ = 0.009). Hence, a simple effects analysis was performed. The results of pairwise comparisons indicate that the antibody levels in time units were affected differently depending on prior COVID-19 infection. For those who reported prior COVID-19 infection, significant differences were revealed between ≤2 months and 2–4 months, as well as at 4+ months (all *p*-values < 0.001), but there was no significant difference between 2–4 months and 4+ months (*p*-value = 0.215). In the subgroup without COVID-19 infection, significant differences were obtained between ≤2 months and 2–4 months, and between 2–4 months and 4+ months (*p*-value < 0.001), but there were no differences in mean antibody titers between ≤2 months and 4+ months after the full vaccination (*p*-value = 0.225). Finally, there were significant differences for all time units after full vaccination between subjects with and without COVID-19 prior to infection (*p*-values < 0.007): for all time intervals, the antibody titers were significantly lower when no COVID-19 infection was reported (see Figure 6B and Table 9).

### 3.8. Changes in Antibody Levels over 8 Months after Vaccination with JNJ-78436735

The J&J vaccine was the least commonly chosen for injection, especially among “white staff”; hence, it has the fewest observations in the analysis. There were 369 subjects vaccinated with two or three doses of JNJ-78436735. First, eight outliers exceeding the upper quantile by three interquartile ranges in groups defined by the time elapsed after full vaccination (≤2 months, 2–4 months, 4+ months) and COVID-19 prior infection (no, yes) were rejected (see Table 9). Finally, the data counted 361 observations.

There was a significant main effect of time elapsed after full vaccination (*F*(2, 355) = 9.386, *p*-value < 0.001, η^2^ = 0.050). The post hoc tests show that the antibody level at ≤2 months was significantly higher than at 2–4 months after vaccination (*p*-value < 0.001), which is shown in Figure 7A and Table 6.

The main effect of COVID-19 was not significant (*F*(1, 355) = 0.757, *p*-value = 0.385); however, there was a significant interaction between the time elapsed after vaccination and prior COVID-19 infection (*F*(2, 355) = 7.830, *p*-value < 0.001, η^2^ = 0.042). Pairwise comparisons indicate that the antibody level in time units was affected differently depending on prior COVID-19 infection. For subjects who reported prior COVID-19 infection, the antibody titer was significantly higher for ≤2 months than for 2–4 months, as well as 4+ months (*p*-values < 0.001 and *p*-value = 0.003, respectively). There was no significant difference in antibody titer between 2–4 and 4+ months (*p*-value ≈ 1), and no significant differences in times were seen for subjects without COVID-19 prior infection (*p*-values > 0.118). This is illustrated in Figure 7B and Table 9.

## 4. Discussion

Since the start of the COVID-19 pandemic, one of the greatest hopes for society has been vaccines, which are now being used around the world in mass vaccination programs. Despite unclear issues, such as the duration of immunity, the prevention of transmission, and protection against new emerging virus variants, COVID-19 vaccines may indeed be a major instrument in preventing SARS-CoV-2 infection, severe disease, death, and, more generally, in fighting a pandemic. The aim of the presented research was to analyze the dynamics of changes in the concentration of anti-SARS-CoV-2 IgG against the coronavirus S protein, in response to injection with four available vaccines in Poland: BNT162b2, mRNA-1273, ChAdOx1 nCoV-2019, and Ad26.COV2.S. The results presented show values at various time intervals up to one year after full vaccination. While all available vaccines, for which Phase 3 efficacy data are available, rely on the whole viral spike protein as an antigen, its presentation to the immune system varies. Moreover, they vary in their degree of purity, and in the presence of other ingredients that can influence the immune response and trigger adverse events [14].

The first observations showed significantly higher antibody titers in people injected with mRNA vaccines within the first 4 months of full vaccination, compared to vector vaccines. Our previous result (Tretyn et al. 2021 and Szczepanek et al. 2022) as well as those obtained by other researchers also confirm these observations [13,15,16,17]. It has also been shown that adenovirus-based vaccines are limited, in that they induce strong T-cell responses but are less effective at producing neutralizing antibodies [18,19,20,21]. Since the induction of a humoral response allows us to quickly limit viral replication, preventing reinfection and the development of disease symptoms, the vaccine-induced cellular response and immune memory protect against severe disease and hospitalization. The induction of a weaker humoral response after vector vaccination suggests the need for a booster dose, especially in the context of protection against SARS-CoV-2 infection [20]. In addition, due to this fact, people who have received vector vaccines are encouraged to take a booster dose with an mRNA vaccine. According to the latest reports, the administration of heterologous vaccines causes the strongest immune response, compared to injection with homologous vaccines [18,21,22,23].

Comparing the intensity of anti-SARS-CoV-2 antibody production between mRNA vaccines, higher IgG levels were observed in the case of mRNA-1273, in both healthy people before vaccination and those suffering from COVID-19. The results of other experiments also confirm the much higher immunogenicity, and the induction of a greater antibody response by the mRNA-1273 compared to the BNT162b2 vaccine [21,24]. Al-Sadeq et al. compared antibodies’ immune responses between the two mRNA vaccines in seronegative and seropositive individuals. The results show that in the healthy vaccinated group, the mRNA-1327 vaccine induces significantly higher levels of S-RBD total antibodies (3.5-fold; *p* < 0.001), as well as S-RBD IgG (2-fold; *p* < 0.01) and S-IgA (2.1-fold, *p* < 0.001), than the BNT162b2 vaccine. The previously infected group produced a higher level of S-RBD IgG than the healthy-BNT162b2 (*p* = 0.05), but not more than the healthy mRNA-1273 (*p* = 0.9) group [25]. Moreover, Richards et al., when comparing the levels of IgG antibodies for SARS-CoV-2 RBD after injection with the abovementioned two vaccines, observed lower levels with the use of BNT162b2 compared to mRNA-1273 (geometric means, preboost blood draw: 19.1 μg/mL vs. 5.9 μg/mL and postboost blood draw: 68.5 vs. 45.9 μg/mL) [26]. These observations can be explained by the differences in the production processes of these vaccines from Pfizer and Moderna. There are slight differences between the two vaccines, both in terms of RNA carriers, LNP, and subtle product-specific RNA sequence variations [14]. Importantly, 100 μg of RNA was used in the mRNA-1273 vaccine, while only 30 μg in the BNT162b2 [14,27,28]. It is also worth mentioning the slightly longer break between taking the second dose of the vaccine in the case of mRNA-1273. According to the literature data, prolonging the time before taking the second dose has a positive effect on the vaccine immune response [29].

In summarizing SARS-CoV-2 IgG levels over time, after injection with four different vaccines, the antibodies produced after the injection of the mRNA vaccine, especially the Comirnaty, show much greater dynamics of changes. Erice et al. compared the decrease in antibody levels 1.5 months and 3 months after receiving two doses of the BNT162b2 vaccine. The median titers of anti-RBD antibodies were 9356 and 3952 AU/mL, respectively, which also confirms the significant dynamics of changes in antibody levels over time. The high dynamics of antibody changes over time after the Pfizer vaccine were also indicated by Levin et al., comparing the antibody titers at the 6th month after taking the second dose [30]. In addition, large changes in the level of spike RBD-specific antibodies within 4 months from the second dose of mRNA-1237 were observed by Brisotto et al. [31]. Greater stability in the level of antibodies against the SARS-CoV-2 S protein was observed with the ChAdOx1 nCoV-2019 vector vaccine. Other researchers indicate similar observations. Mishra et al. measured anti-SARS-CoV-2 antibody titers 1 month after receiving the first dose of Oxford/AstraZeneca vaccine (Round 1), 1 and 6 months after trimming the second dose (Rounds 2 and 3). The geometric mean IgG titers were 138.01 BAU/mL in Round 1, 176.48 BAU/mL in Round 2, and 112.95 BAU/mL in Round 3 [32].

It has also been noted that vaccinated convalescents maintain high anti-SARS-CoV-2 IgG titers for much longer, and these are approximately 2–10 times higher compared to seronegative vaccinates at the same time. In the case of the ChAdOx1 nCoV-2019 vector vaccine, there are differences between pre-vaccination seronegative and seropositive persons, although they are not as high as for other vaccines. Studies on the persistence of anti-SARS-CoV-2 IgG in people who are not vaccinated after COVID-19 show that in about 80% of individuals, they persist as long as 13 months after the infection [33,34,35]. Moreover, they also confirm significantly lower decreases in IgG and neutralizing antibodies in people vaccinated with a previous COVID-19 infection, compared to those only vaccinated [36]. Eyre et al. also compared levels of anti-SARS-CoV-2 antibodies in patients with and without prior infection. Quantitative antibody responses were higher after previous infection: median >21 days post-first Pfizer/BioNTech 14,604 AU/mL vs. 1028 AU/mL without prior infection. Individuals vaccinated with the Oxford/AstraZeneca vaccine had lower readings post-first dose than Pfizer/BioNTech recipients, with and without previous infection—10,095 and 435 AU/mL, respectively. The antibody responses at >21 days after second Pfizer vaccination in those not previously infected, 10,058 AU/mL, were similar to those after prior infection followed by one vaccine dose [37]. Angyal et al. also studied 289 healthcare workers after vaccination with BNT162b2, and indicated significantly higher levels of anti-SARS-CoV-2 antibodies in the group of people who had COVID-19 before vaccination, compared to those who were seronegative [38].

In the range of 5–8 months after full vaccination, the levels of anti-SARS-CoV-2 IgG after injection with equal vaccines were similar and reasonably stable. The exception is the JNJ-78436735, for which the IgG levels were significantly lower than in those vaccinated with other preparations, or were undetectable. Unfortunately, there are no reports in the literature presenting the results of antibody levels in the range of 6–8 months after taking the second dose of vaccination, especially those comparing the four vaccines tested by us. Achiron et al. presented anti-SARS-CoV-2 IgG levels after vaccination with BNT162b2 at intervals of 2–4, 4–6, and 6–8 months after the second dose. Contrary to our observations, the median antibody concentration was significantly lower in the 6–8 month time frame compared to the 4–6 month time frame [39]. There is no doubt that the JNJ-78436735 vaccine induced the weakest immune response among all four vaccines available in Poland. For example, van Gils et al. compared the humoral response 4 weeks after full injection with four COVID-19 vaccines. The titers of S protein binding and neutralizing antibodies were higher in individuals after vaccination with BNT162b2 and mRNA-123 (median MFI titers 1482 and 1257, respectively; median ID_50_ titers of 3061 and 1891, respectively), and substantially lower in those vaccinated with the adenovirus vector-based vaccines ChAdOx1 nCoV-2019 and Ad26.COV2.S (median MFI titers 74 and 52, respectively; median ID_50_ titers of 241 and 119, respectively) [40]. Apart from the fact that it is a vectored vaccine, one of the main reasons for this is that it was given in a single-dose, which is probably insufficient to obtain a strong effect. In conducting the research, however, we observed that vaccination with the Janssen/Johnson & Johnson preparation in convalescents induced a high production of anti-SARS-CoV-2 IgG antibodies [40].

In the 9th month of monitoring the level of anti-SARS-CoV-2 IgG, we observed significant increases in participants vaccinated with BNT162b2 and mRNA-1327. This increase was caused by the intake of a booster dose by some of the health workers. The second group of individuals characterized by a high level of antibodies comprised those who were vaccinated in the “0” group and did not declare taking a booster dose. It is likely that this group of participants had COVID-19 asymptomatically, and were unaware of it. This fact confirms the effectiveness of vaccinations in relieving the symptoms of COVD-19 and protecting against the severe course of the disease. On the other hand, it allows one to realize the huge unconscious transmission of the SARS-CoV-2 virus, especially in the group of healthcare professionals. With this in mind, more frequent screening for SARS-CoV-2 infection in this work group should be considered.

As previously described, the national vaccination system in Poland started at the end of December 2020. In the summer of 2021, the number of SARS-CoV-2 infections began to increase, as vaccine protection against COVID-19 weakened and the new B.1.617.2 (Delta) virus variant spread. This prompted the approval of a booster dose. In Poland, its application began on 24 September 2021, and on 4 November it could be taken by any adult. In analyzing the responses after the second and third doses of BNT162b2, we observed significantly higher levels of anti-SARS-CoV-2 IgG after the booster dose compared to the second dose. Lustig et al. compared the humoral response in healthcare workers after the second and third doses of the BNT162b2 vaccine. The estimated geometric mean titer for IgG following the second dose was 1586 BAU/mL, and for IgG following the third dose, it was 2745 BAU/mL. Thus, a 1.7-fold increase in IgG levels occurred following the third dose in comparison to the second dose [41]. Similarly, Flaxman et al. reported that antibody levels 28 days after the third dose of ChAdOx1 nCoV-19 were significantly higher than at 28 days after the second dose, with median total IgG titers 3746 EUs and 1792 EUs, respectively [29].

An increase in the level of antibodies was observed as early as around the 5th day after the injection of the booster dose, and was maximized around the 14th day. In contrast, with the second dose of BNT162b2, peak antibody levels were observed on days 14–21 post-vaccination [13,17]. Demonbreun et al. also observed a large antibody response 6–10 days after the booster dose. Moreover, anti-SARS-CoV-2 IgG levels exceeded those documented after natural COVID-19 infection after two doses of the vaccine, and after both infection and vaccination [42]. In the case of the booster dose, no significant differences in the level of antibodies were observed between people who had never suffered from COVID-19 and convalescents. The same conclusions were reached in the research by Glück et al. These scientists did not observe differences in the titers of anti-SARS-CoV-2 antibodies in the individuals post-COVID-19 and after the booster dose compared to the vaccinated only group [43].

The third dose induced a significant production of anti-SARS-CoV-2 IgG in people who did not respond to vaccination with adequate antibody production after the first two doses. Moreover, out of 3192 patients who took a booster dose, 3 patients tested negative for anti-SARS-CoV-2 IgG. These were two women, aged 42 and 53, with malignant neoplasms of the lymphatic system, and one 42-year-old patient with thyroid disease. These results confirm the assumption that the administration of three doses of the COVID-19 vaccine offers people with diseases that impair immune response the chance to obtain protection. Additionally, Lustig et al. showed that the third dose of BNT162b2 vaccine produced a stronger response in healthcare workers ≥ 60 years of age, and in those with more than two comorbidities who showed a poorer response after the second dose [41].

## 5. Conclusions

The conducted research allowed us to characterize the dynamics and durability of antibodies against SARS-CoV-2 S1 subunit in response to vaccination with four vaccines available in Poland, in the period from 1 to 12 months after injection. Despite the stronger response elicited by mRNA vaccines, half a year after full vaccination, IgG levels were similar for the BNT162b2, mRNA-1327, and ChAdOx1 nCoV-2019 vaccines. The least immunogenic vaccine was JNJ-78436735. The cut-off date for protection against the disease seems to be the period of 8–9 months from the vaccination for mRNA vaccines and 5–6 months for vector vaccines. This is indicated by the large group of medical staff who presumably underwent asymptomatic SARS-CoV-2 infection. This fact also indicates the need for more frequent checks of healthcare workers for the presence of SARS-CoV-2. The introduction of a booster dose in September 2021 was the right decision, and it could have a real impact on restricting the further transmission of the virus. The low infection rate after full vaccination supports this conclusion. The high antibody titers in a significant number of participants who were vaccinated 6 months prior to the test suggest that these people may have developed asymptomatic SARS-CoV-2 infection. On the one hand, this confirms the effectiveness of vaccines in protecting against severe disease, even 6–12 months after full vaccination. On the other hand, it prompts us to consider whether it would be worth introducing serological tests in people referred for mandatory booster vaccinations. Many healthcare workers now have higher antibody titers as a result of latent but natural immunization than those after the third dose.

## Figures and Tables

**Figure 1 vaccines-10-00506-f001:**
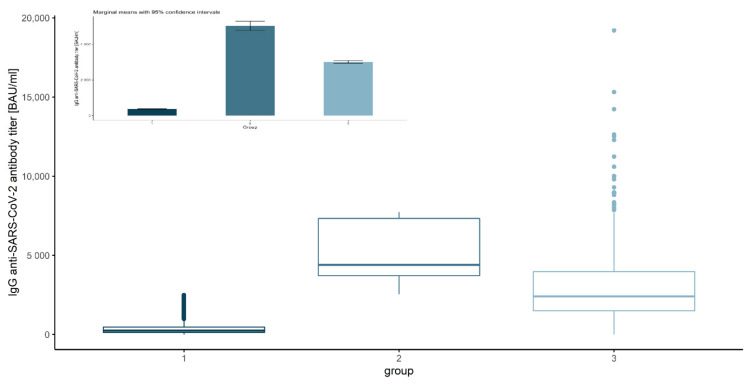
Comparison of anti-SARS-CoV-2 antibody levels by group without distinguishing among vaccines. Group 1 (dark blue box): 2 doses more than 6 months before measurement, low antibody level (<2500), regardless of the declaration of COVID-19 prior infection; group 2 (blue box): 2 doses more than 6 months before measurement, high antibody levels (>2500), no declaration of COVID-19 prior to infection; group 3 (light blue box): 3 doses, regardless of the declaration of COVID-19 prior to infection and antibody levels.

**Figure 2 vaccines-10-00506-f002:**
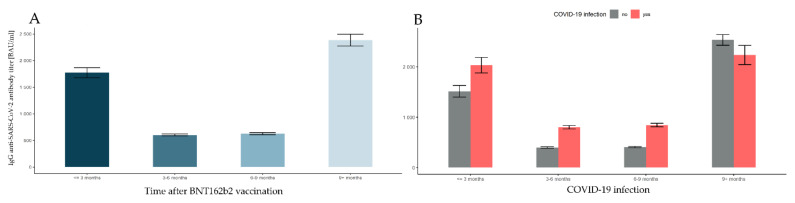
Comparisons of antibody levels following full vaccination with BNT162b2, taking into account the time factor and the incidence of SARS-CoV-2 infection. The error bars for marginal means with 95% confidence intervals (bootstrapped) are presented in turn: (**A**) the time elapsed after the full BNT162b2 vaccination; (**B**) the prior COVID-19 infection grouped with different time intervals elapsed after full BNT162b2 vaccination.

**Figure 3 vaccines-10-00506-f003:**
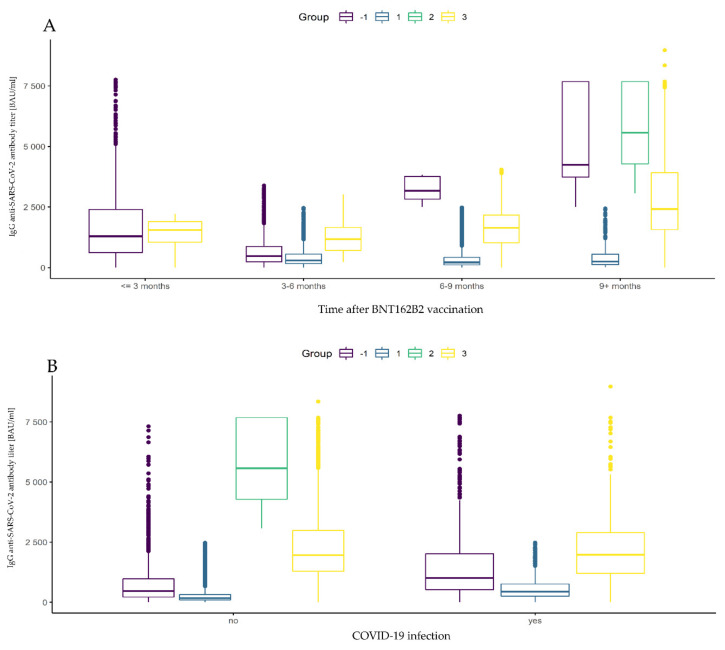
Comparison of antibody levels after full BNT162b2 vaccination by group according to the time factor and frequency of SARS-CoV-2 infection. Box and whisker plots are presented in turn: (**A**) the time elapsed after the full BNT162b2 vaccination; (**B**) evidence of SARS-CoV-2 infection before vaccination. Group 1 (dark blue box): 2 doses, more than 6 months before measurement, low antibody levels (<2500 BAU/mL), regardless of the declaration of prior COVID-19 infection. Group 2 (green box): 2 doses, more than 6 months before the measurement, high antibody levels (>2500 BAU/mL), no declaration of prior COVID-19 infection. Group 3 (yellow box): 3 doses, regardless of the declaration of prior COVID-19 infection and antibody level. Group 1 (purple box): a group of participants who did not meet the criteria for belonging to Groups 1, 2, or 3.

**Figure 4 vaccines-10-00506-f004:**
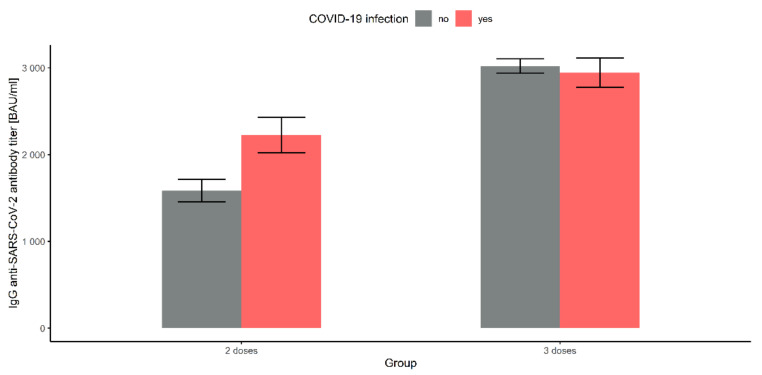
Comparison of the mean antibody levels over 3 months after vaccination with 2 and 3 doses of BNT162b2.

**Figure 5 vaccines-10-00506-f005:**
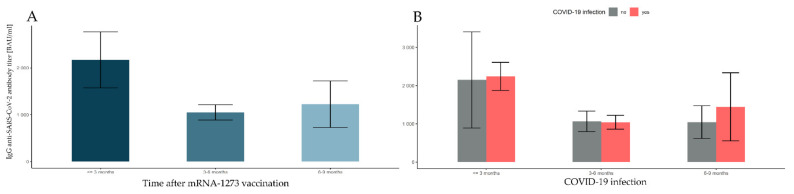
Comparisons of antibody levels following full vaccination with mRNA-1273, taking into account the time factor and the incidence of SARS-CoV-2 infection. The error bars for marginal means with 95% confidence intervals (bootstrapped) are presented in turn: (**A**) the time elapsed after the full mRNA-1273 vaccination; (**B**) the prior COVID-19 infection group with different time intervals elapsed after full mRNA-1273 vaccination.

**Figure 6 vaccines-10-00506-f006:**
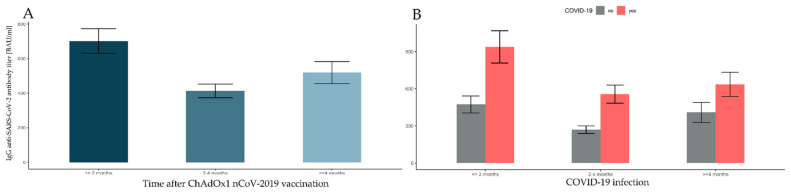
Comparisons of antibody levels following full vaccination with ChAdOx1 nCoV-2019, taking into account the time factor and the incidence of SARS-CoV-2 infection. The error bars for marginal means with 95% confidence intervals (bootstrapped) are presented in turn: (**A**) the time elapsed after the full ChAdOx1 nCoV-2019 vaccination; (**B**) prior COVID-19 infection grouped with different time intervals elapsed after full ChAdOx1 nCoV-2019 vaccination.

**Figure 7 vaccines-10-00506-f007:**
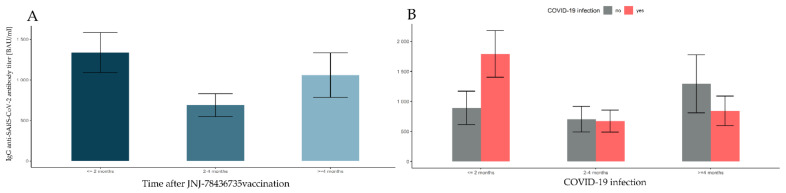
Comparisons of antibody levels following full vaccination with JNJ-78436735, taking into account the time factor and the incidence of SARS-CoV-2 infection. The error bars for marginal means with 95% confidence intervals (bootstrapped) are presented in turn: (**A**) the time elapsed after the full JNJ-78436735 vaccination; (**B**) prior COVID-19 infection grouped with different time intervals elapsed after full JNJ-78436735 vaccination.

**Table 1 vaccines-10-00506-t001:** Characteristics of the study group.

		Total	Unvaccinated	BNT162b2	mRNA-1273	ChAdOx1 nCoV-2019	Ad26.COV2.S
Age (y)	Under 35	3337 (17.9%)	383 (24.4%)	2539 (16.5%)	222 (22.8%)	106 (37.9%)	87 (23.6%)
	36–45	3720 (20.0%)	354 (22.5%)	3020 (19.6%)	210 (21.6%)	73 (26.1%)	63 (17.1%)
	46–55	6201 (33.3%)	459 (29.2%)	5247 (34.0%)	291 (29.9%)	59 (21.1%)	145 (39.3%)
	56–65	4622 (24.8%)	341 (21.7%)	3960 (25.7%)	222 (22.8%)	33 (11.8%)	66 (17.9%)
	66+	729 (3.9%)	33 (2.1%)	650 (4.2%)	29 (3.0%)	9 (3.2%)	8 (2.2%)
Gender	Female	15,133 (81.3%)	1308 (83.3%)	12527 (81.3%)	784 (80.5%)	225 (80.4%)	289 (78.3%)
	Male	3477 (18.7%)	262 (16.7%)	2890 (18.7%)	190 (19.5%)	55 (19.6%)	80 (21.7%)
COVID-19 status	No infection	12,397 (66.6%)	941 (60.1%)	10550 (68.4%)	553 (56.8%)	156 (55.7%)	194 (52.6%)
	Before vaccination	6188 (33.3%)	621 (39.6%)	4848 (31.4%)	421 (43.2%)	123 (43.9%)	175 (47.4%)
	After vaccination	575 (3.1%)	0 (0.0%)	554 (3.6%)	13 (1.3%)	0 (0.0%)	8 (2.2%)
	Twice	19 (0.1%)	5 (0.3%)	14 (0.1%)	0 (0.0%)	0 (0.0%)	0 (0.0%)
	Potential	266 (1.4%)	15 (1.0%)	221 (1.4%)	16 (1.6%)	11 (3.9%)	3 (0.8%)

**Table 2 vaccines-10-00506-t002:** Minimal values and the numbers of outliers for each group determined by the time after vaccination and COVID-19 prior infection for BNT162b2 and mRNA-1273.

	Time after Vaccination	≤3 Months		3–6 Months		6–9 Months		9+ Months		Total Number of Outliers
	COVID-19 Prior to Infection	No	Yes	No	Yes	No	Yes	No	Yes	
**BNT162b2**	Value	7680	8622	1880	3442.4	2524	4091.2	19,200	12,625	982
	Number of observations	5	8	99	42	679	146	1	2	
**mRNA-1273**	Value	6957.6	13,977.5	7680	4412.6	-	-	-	-	11
	Number of observations	5	1	1	4	0	0	-	-	

**Table 3 vaccines-10-00506-t003:** Minimal values and the numbers of outliers for each group determined by the time after vaccination and COVID-19 prior infection for ChAdOx1 nCoV-2019 and JNJ-78436735.

	Time after Vaccination	≤2 Months		2–4 Months		4+ Months		Total Number of Outliers
	COVID-19 Prior to Infection	No	Yes	No	Yes	No	Yes	
**ChAdOx1 nCoV-2019**	Value	1914.1	4117	2188.4	3011	3124.6	3484.6	57
	Number of observations	3	1	2	5	33	13	
**JNJ-78436735**	Value	6441.8	-	-	3561.4	-	7268.4	8
	Number of observations	2	0	0	5	0	1	

**Table 4 vaccines-10-00506-t004:** Descriptive statistics for healthy unvaccinated subjects.

Group	Minimum	1st Quartile	Median	Mean	3rd Quartile	Maximum	SD
Overall (*N* = 941)	3.2	3.2	13.96	105.42	94.75	7680	488.80
No antibodies (≤35.2)	3.2	3.2	3.2	6.959	5.71	35.09	7.44
Antibodies detected (>35.2)	35.87	69.01	119.30	247.62	230.93	7680	741.95

**Table 5 vaccines-10-00506-t005:** Comparison of mean anti-SARS-CoV-2 IgG antibody titers among vaccines in the months following full vaccination.

Month	Pfizer/BioNTech (BNT162b2)		Moderna (mRNA-1273)		Oxford/AstraZeneca (ChAdOx1 nCoV-2019)		Janssen/Johnson & Johnson (Ad26.COV2.S.)	
	SARS-CoV-2 Infection							
	No	Yes	No	Yes	No	Yes	No	Yes
	Mean Anti-SARS-CoV-2 IgG Concentration (BAU/mL) SD; 95%CI							
1	**3004.96** 1858.58; 3421.26-2588.66	**4358.83** 3386.95; 5274.45–3443.21	**4405.68** 2504.75; 5997.13–2814.24	**8079.23** 3690.02; 11491.93–4666.53	**706.5** 673.53; 882.02–530.98	**1065.56** 818.68; 1308.68–822.44	**1022.15** 1807.59; 1643.08-401.22	**2085.63** 1448.83; 2697.42–1473.84
2	**2067.00** 1720.81; 2440.44–1693.56	**3022.72** 2601.41; 3629.67–2415.77	**3537.75** 2779.49; 7960.54–885.04	**4398.72** 2650.63; 6179.43–2618	**526.37** 448.52; 725.23–327.51	**1136.3** 856.56; 1497.99–774.61	**1208.41** 1789; 1813.72–603.1	**2300.33** 2385.00; 3225.14–1375.52
3	**1342.04** 1105; 1513.49–1170.59	**1701.88** 1111.18; 1876.49–1527.27	**1643.6** 1492.43; 2505.31–781.89	**2551.25** 1502.58; 3459.25–1643.25	**287.26** 249.52; 362.22–212.3	**790.46** 709.42; 986–594.91	**720.98** 957.9; 1135.21–306.75	**959.45** 852.71; 1290.1–628.8
4	**975.90** 919.26; 1108.88–842.92	**1590.61** 1463.17; 1822.78–1358.44	**1460.96** 968.00; 1748.42–1173.5	**1230.23** 702.04; 1568.6–891.86	**271.72** 279.64; 326.93–216.52	**586.75** 582.35; 716.35–457.16	**745.59** 925.35; 1041.53–449.64	**1120.15** 1565.49; 1666.37–573.92
5	**627.01** 688.93; 686.19–567.83	**1180.34** 1375.07; 1323.06–1037.62	**1241.54** 800.07; 1725.02–758.06	**1893.14** 1715.2; 2719.84–1066.44	**328.80** * 59.48; 424.49–233.11	**893.36** * 1671.5; 1283.35–503.37	**634.46** * 940.99; 985.83–283.09	**804.31** * 1037.53; 1178.38–430.24
6	**543.20** 722.77; 600.77–485.64	**871.58** 761.17; 947.56–795.61	**1038.70** 1095.82; 1537.52–539.89	**1359.86** 1334.91; 1937.12–782.6	**1002.52** 2177.32; 1419.83–585.2	**855.43** 1241.35; 1181.82–529.03	**1080.98** 1335.62; 1820.62–341.34	**1345.10** 1811.25; 2390.88–299.32
7	**483.31** 875.3; 535.72–430.9	**811.35** 955.99; 885.85–736.85	**1120.12** 1612.27; 1834.97–405.28	**760.04** 338.71; 955.6–564.47	**1384.63** 2247.99; 1799.89–969.36	**1402.78** 2030.03; 1886.83–918.74	**1535.15** 1575.86; 2662.46–407.85	**790.33** 701.39; 1195.29–385.36
8	**328.62** 613.31; 352.95–304.3	**707.54** 896.02; 762.81–652.27	**1052.66** 1242.57; 1691.53–413.79	**691.53** 474.96; 1447.29–64.24	**348.94** 468.06; 781.82–83.94	**422.00** 197.30; 586.95–257.05	**1671.23** 1111.52; 3439.9–97.45	
9	**753.22** * 1419.24; 811.01–695.44	**1109.24** * 1698.73; 1217.4–1001.08	**2294.25** * 1940.98; 5382.78–794.28	**908.6** * 695.57; 1772.26–44.94				
10	**1952.48** 2126.39; 2061.6–1843.35	**2155.8** 2271.71; 2346.77–1964.83	**242.95** 60.6; 787.41–301.51	**1374.9** 236.32; 3498.1–748.31				
11	**2549.28** 2242.13; 2662.16–2436.4	**2284.43** 2150.25; 2490.67–2078.19						
12	**2391.13** 3100.24; 7324.3–2542.05							

* Starting vaccination with a booster dose; the beginning of the “fourth wave” infections.

**Table 6 vaccines-10-00506-t006:** A summary of the antibody levels for healthy vaccinated subjects.

Group		Minimum	1st Quartile	Median	Mean	3rd Quartile	Maximum	SD
Healthy vaccinated (2 dose) more than 6 months before the antibody measurement	Overall (*N =* 6620)	3.2	101.4	185.2	424.6	354.1	7730	911.40
	Low antibody level (≤2500)	3.2	95.85	169.85	275.08	306.30	2473.60	337.91
	High antibody level (>2500)	2539	3763	4412	5062	7159	7730	1793.01
Vaccinated (3 dose), overall		3.2	1490	2405.80	3007.80	3971.6	19,200	2161.04

**Table 7 vaccines-10-00506-t007:** Estimated marginal means, the 95%-confidence intervals, and the numbers of observations for each group determined by levels of prior COVID-19 infection, the time after vaccination, and its interaction for BNT162b2 and mRNA-1273.

Factors and Levels		BNT162b2		mRNA-1273	
		Mean * (Number of Observations)	95% CI	Mean * (Number of Observations)	95% CI
**COVID-19**	no	1216.062 (9687)	(1176.006, 1256.118)	1375.350 (150)	(967.256, 1783.444)
	yes	1478.892 (4568)	(1415.960, 1541.824)	1571.633 (112)	(1238.939, 1904.328)
**Time after vaccination**	≤3 months	1772.822 (948)	(1678.493, 1867.151)	2170.207 (121)	(1573.818, 2766.596)
	3–6 months	601.301 (3461)	(582.563, 620.039)	1050.766 (107)	(889.980, 1211.552)
	6–9 months	626.441 (7909)	(607.381, 645.500)	1224.907 (34)	(727.943, 1721.871)
	9+ months	2384.526 (1937)	(2274.893, 2494.159)	-	-
**Time after vaccination and COVID-19 infection**	≤3 months, no	1517.258 (512)	(1401.604, 1632.913)	2149.050 (72)	(891.700, 3406.400)
	≤3 months, yes	2034.188 (436)	(1880.238, 2188.138)	2240.479 (49)	(1873.151, 2607.806)
	3–6 months, no	399.911 (2109)	(384.389, 415.432)	1065.031 (55)	(796.358, 1333.704)
	3–6 months, yes	803.615 (1352)	(769.536, 837.694)	1041.315 (52)	(860.521, 1222.110)
	6–9 months, no	407.1315 (5546)	(393.041, 421.222)	1044.596 (23)	(617.441, 1471.751)
	6–9 months, yes	845.875 (2363)	(810.85, 880.900)	1445.786 (11)	(555.032, 2336.539)
	9+ months, no	2537.310 (1520)	(2431.067, 2643.552)	-	-
	9+ months, yes	2237.735 (417)	(2046.920, 2428.549)	-	-

* Mean IgG concentration in BAU/mL.

**Table 8 vaccines-10-00506-t008:** 95% confidence intervals (lower and upper bounds) for the mean level of antibodies in each of the three groups.

Group	Mean	Lower CI	Upper CI
1	374.50	366.35	382.64
2	5034.11	4777.52	5290.70
3	3007.76	2932.75	3082.77

**Table 9 vaccines-10-00506-t009:** Estimated marginal means, the 95% confidence intervals, and the numbers of observations for each group determined by levels of COVID-19 prior infection, the time after vaccination, and the interaction for ChAdOx1 nCoV-2019 and JNJ-78436735.

Factors and Levels		ChAdOx1 nCoV-2019		JNJ-78436735	
		Mean * (Number of Observations)	95% CI	Mean * (Number of Observations)	95% CI
**COVID-19**	no	382.500 (507)	(346.726, 418.274)	960.268 (192)	(765.939, 1154.597)
	yes	706.674 (395)	(648.005, 765.343)	1099.011 (169)	(936.426, 1261.595)
**Time after vaccination**	≤2 months	702.293 (245)	(631.245, 773.340)	1337.550 (172)	(1090.454, 1584.645)
	2–4 months	413.318 (336)	(373.365, 453.27)	688.845 (131)	(547.739, 829.95)
	4+ months	519.249 (321)	(455.447, 583.051)	1058.729 (58)	(784.584, 1332.875)
**Time after vaccination and COVID-19 infection**	≤2 months, no	473.535 (123)	(404.932, 542.138)	891.668 (92)	(612.647, 1170.689)
	≤2 months, yes	938.1125 (122)	(807.835, 1068.390)	1791.968 (80)	(1402.612, 2181.325)
	2–4 months, no	269.658 (188)	(238.669, 300.647)	704.624 (70)	(492.279, 916.969)
	2–4 months, yes	556.898 (148)	(483.554, 630.242)	672.644 (61)	(488.824, 856.464)
	4+ months, no	409.152 (196)	(328.539, 489.765)	1293.385 (30)	(810.096, 1776.675)
	4+ months, yes	634.559 (125)	(536.503, 732.615)	842.969 (28)	(596.795, 1089.142)

* Mean IgG concentration in BAU/mL.

## Data Availability

The data presented in this study are available upon request from the corresponding author. The data are not publicly available due to their containing information that could compromise the privacy of research participants.

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
