# Peer review of "The Dynamics of Changes in the Concentration of IgG against the S1 Subunit in Polish Healthcare Workers in the Period from 1 to 12 Months after Injection, Including Four COVID-19 Vaccines"

_vaccines, 2022, doi:10.3390/vaccines10040506_

Round 1

Reviewer 1 Report

I would like to thank you for conducting and reporting on the important information that showed the antibody induction resulted from vaccine against COVID-19 in a very easy-to-understand manner.

A lot of data indicate that vaccination does not block the infection, and Table 7 also shows the COVID-19 status. If the vaccination suppresses mortality and severe COVID, it needs to show that data as well as COVID-19 status.

Do the daily death in Poland shown in worldometers.info (https://www.worldometers.info/coronavirus/country/poland/) tell that the vaccine is controlling the number of deaths?

Minor revision; The title should state that the survey was performed in Poland.

Author Response

I would like to thank you for conducting and reporting on the important information
that showed the antibody induction resulted from vaccine against COVID-19 in a
very easy-to-understand manner.

Thank you very much for these words. We are very pleased that our work has been
appreciated.

A lot of data indicate that vaccination does not block the infection, and Table 7 also
shows the COVID-19 status. If the vaccination suppresses mortality and severe
COVID, it needs to show that data as well as COVID-19 status.

Do the daily death in Poland shown in worldometers.info
(https://www.worldometers.info/coronavirus/country/poland/) tell that the vaccine is
controlling the number of deaths?

Unfortunately, we have not analyzed the mortality and relieving of COVID-19 and do not
have such data. However, the mortality rate in Poland is really very high. This is mainly due
to the low percentage of people vaccinated in Poland, amounting to only 58% of the
population today. However, there is no doubt that vaccination has a very high degree of
protection against severe COVID-19 and death. The report of experts of the National Institute
of Public Health presents preliminary data on mortality due to COVID-19 in the population
of unvaccinated and vaccinated against COVID-19 in Poland. Higher rates of COVID-19
mortality among the unvaccinated compared to the vaccinated were observed in all age
groups, including > 138 times greater among unvaccinated people aged 61-70 years and 126
times greater among unvaccinated people aged 71-80 years. In the event of a severe course of
COVID-19 or the death of the vaccinated person, comorbidities should also be taken into
account, which are very often of decisive importance.

Minor revision; The title should state that the survey was performed in Poland.

Thank you very much for your valuable remark. The title has been changed.

Reviewer 2 Report

Dear Authors,

It was my pleasure to review the manuscript titled The dynamics of changes in the concentration of IgG against the S1 subunit in the period from 1 to 12 months after injection, including four COVID-19 vaccines, a highly detailed and thorough study designed to provide an assessment as to the dynamics and durability of anti-SARS-CoV-2 antibodies (Immunoglobulin G targeting SARS-CoV 2 spike protein S1 subunit) elicited by the four vaccines then available in Poland, in a time frame encompassing 1 to 12 months following administration.

The article's main objective is clearly stated and its chief strength resides in its relevance and underlying merit in terms of elaborating on the role and value of vaccination to put an end to the pandemic, at least in its deadly manifestations.
The methodology and analytical standards are accurately expounded upon and rather solid, (the through enzyme-linked immunosorbent assay is mentioned only as an acronym, please add the full definition), as far as I was able to determine.
The tables are well-constructed and valuable when it comes to conveying core messages and measurements at the heart of the article's objective.
The authors have included the S1 subunit, which contains a receptor-binding domain (RBD) that recognizes and binds to the host receptor angiotensin-converting enzyme 2 (ACE2), in the title but it would be worth drawing, in my view, a distinction with the S2 subunit,  which is known to mediate viral cell membrane fusion.

The article has major weaknesses having to do with its objectionable structure, placing the results following the introduction, then discussion, materials and methods and conclusions.
For the sake of clarity and readability, I would recommend that the authors restructure their manuscript following the sequence: Introduction, Materials (NOT "Material", as wrongly spelled in the abstract), Results and Discussion, Conclusions.
Given the degree of complexity within the core analysis undertaken by the authors, restructuring and streamlining the overall presentation is simply non-negotiable.
The article is poorly written and needs thorough proof-reading by a native speaker of English in order to fix its often  syntax and grammar. Typos and misspellings are also rife throughout.

Some instances:

Line 66: In Poland, the BNT162b2 is available from December 27, 2020. (Wrong tense)

Line 107: As the times at which the highest immunological parameters are obtained in response to vaccination against SARS-CoV-2 have already been characterized, the study of the protective effect is still ongoing (please rephrase or split up).

Line 325: There were 959 subjects with 2 or 3 dose AstraZeneca vaccinated (rephrase). 

Line 561: It is carried out from 01/03/2020 to 30/06/2022, in partnership between the Kuyavian-Pomeranian Voivodeship Self-government and 29 medical entities from the entire voivodeship. (If it is still ongoing, fix the verb tense).

I am looking forward to reviewing a new improved version of the article as the authors and the Editor see fit.

Sincerely,

Author Response

Reviewer 2

It was my pleasure to review the manuscript titled The dynamics of changes in the concentration of IgG against the S1 subunit in the period from 1 to 12 months after injection, including four COVID-19 vaccines, a highly detailed and thorough study designed to provide an assessment as to the dynamics and durability of anti-SARS-CoV-2 antibodies (Immunoglobulin G targeting SARS-CoV 2 spike protein S1 subunit) elicited by the four vaccines then available in Poland, in a time frame encompassing 1 to 12 months following administration.

The article's main objective is clearly stated and its chief strength resides in its relevance and underlying merit in terms of elaborating on the role and value of vaccination to put an end to the pandemic, at least in its deadly manifestations.

Thank you very much for this comment. We are glad that our work has been appreciated in such important aspects for us.

The methodology and analytical standards are accurately expounded upon and rather solid, (the through enzyme-linked immunosorbent assay is mentioned only as an acronym, please add the full definition), as far as I was able to determine.

Indeed, the development of the acronym has been neglected. This has been completed in Materials and Methods, subchapter 2.2. IgG anti-SARS-CoV-2 assays (lines 181-182). The acronym has been left in the abstract.

The tables are well-constructed and valuable when it comes to conveying core messages and measurements at the heart of the article's objective.

The authors have included the S1 subunit, which contains a receptor-binding domain (RBD) that recognizes and binds to the host receptor angiotensin-converting enzyme 2 (ACE2), in the title but it would be worth drawing, in my view, a distinction with the S2 subunit,  which is known to mediate viral cell membrane fusion.

Thank you for this advice. We did not make a drawing, but in the introduction we added information about the structure of the S protein, the function of individual subunits and the importance of the S1 subunit (lines 109-124). We hope you will find it satisfactory.

The article has major weaknesses having to do with its objectionable structure, placing the results following the introduction, then discussion, materials and methods and conclusions.

For the sake of clarity and readability, I would recommend that the authors restructure their manuscript following the sequence: Introduction, Materials (NOT "Material", as wrongly spelled in the abstract), Results and Discussion, Conclusions.

Given the degree of complexity within the core analysis undertaken by the authors, restructuring and streamlining the overall presentation is simply non-negotiable.

Thank you very much for your valuable comment. The article has been reorganized as recommended by the reviewer. Without a doubt, it is now more readable and transparent. The error in the Abstract has also been corrected.

The article is poorly written and needs thorough proof-reading by a native speaker of English in order to fix its often  syntax and grammar. Typos and misspellings are also rife throughout.

Thank you very much for checking the manuscript linguistically and for finding a solution. In the first place, the text was corrected by us in terms of incorrectly formulated or unclear tasks. Then the text was subject to professional linguistic proofreading at MDPI English Editing Services (see – ms - tracking changes).

Some instances:

Line 66: In Poland, the BNT162b2 is available from December 27, 2020. (Wrong tense)

The sentence was changed to: “BNT162b2 has been available in Poland from December 27, 2020.” Lines 65-66.

Line 107: As the times at which the highest immunological parameters are obtained in response to vaccination against SARS-CoV-2 have already been characterized, the study of the protective effect is still ongoing (please rephrase or split up).

The sentence was changed to: “On the basis of numerous scientific studies on the short-term vaccine response, the time from full vaccination in which the highest immunological parameters and, thus, the highest protection against infection are achieved was characterized. However, research is still ongoing to determine the duration of this protective period.” Lines 125-128.

Line 325: There were 959 subjects with 2 or 3 dose AstraZeneca vaccinated (rephrase).

The sentence was changed to: “There were 959 participants vaccinated with AstraZeneca.” Line 439.

Line 561: It is carried out from 01/03/2020 to 30/06/2022, in partnership between the Kuyavian-Pomeranian Voivodeship Self-government and 29 medical entities from the entire voivodeship. (If it is still ongoing, fix the verb tense).

The sentence was changed to: “The implementation of the program began on March 1, 2020 and its continuation is planned until June 30, 2022. Twenty-nine medical entities from the entire voivodship cooperate in the partnership between the Kuyavian–Pomeranian Voivodeship Self-Government. The current study takes into account the data obtained during diagnostic tests performed by the end of 2021.” Lines 152-156.

Reviewer 3 Report

In my opinion, the study presents interesting results on COVID-19 vaccination. The research team has concluded that The cut-off date for protection against the disease seems to be the period of 8-9 months from the vaccination for mRNA vaccines and 5-6 months for vector vaccines. The introduction of a booster dose was the right decision that could have a real impact on restricting the further transmission of the virus.
The study is well-designed, and the materials and methods are precisely described. Also, The results and their discussion are well-presented. 

Minor comments:
In the introduction section, I recommend the authors provide additional information about IgG, the S1 subunit, and their role during vaccination. This will help the non-specialized reader understand the aim of the study. Additionally, it would be better to highlight this point in the conclusion section. 
Another important point:
Your previously published paper ref [14] (doi:10.3390/cells10081952) seems to have common findings with your current study. Could you please explain this point and highlight the difference between the currently acquired results and what you have previously published? This explanation is highly recommended to be highlighted in the current study. 
Finally, I recommend the authors double-check the full text for grammatical and typing errors. 

Author Response

Reviewer 3

In my opinion, the study presents interesting results on COVID-19 vaccination. The research team has concluded that The cut-off date for protection against the disease seems to be the period of 8-9 months from the vaccination for mRNA vaccines and 5-6 months for vector vaccines. The introduction of a booster dose was the right decision that could have a real impact on restricting the further transmission of the virus.

The study is well-designed, and the materials and methods are precisely described. Also, The results and their discussion are well-presented.

Thank you for your interest and appreciation of our work. We are very pleased.

Minor comments:

In the introduction section, I recommend the authors provide additional information about IgG, the S1 subunit, and their role during vaccination. This will help the non-specialized reader understand the aim of the study. Additionally, it would be better to highlight this point in the conclusion section.

Thank you for this advice. It certainly increased the value of the manuscript. The introduction has been supplemented with information on the important role of the S1 subunit. Lines 109-124. Moreover, information was added in the summary that the research was based on the detection of this subunit. Line 673.

Another important point:

Your previously published paper ref [14] (doi:10.3390/cells10081952) seems to have common findings with your current study. Could you please explain this point and highlight the difference between the currently acquired results and what you have previously published? This explanation is highly recommended to be highlighted in the current study.

The presented research is a continuation of the research carried out on the analysis of the responses to vaccination against COVID-19. Two of our works have been published so far: Tretyn et al. 2021 (doi: 10.3390 / cells10081952) and Szczepanek et al. 2022 (doi.org/10.3390/vaccines10010099). Despite the fact that these works are related to similar topics, they differ significantly in many aspects, which we will briefly describe.

The aim of the study by Tretyn et al. 2021 was to assess the titer of anti-SARS-CoV-2 IgG against the S1 subunit of the viral spike protein as a marker of the humoral response in 477 adult volunteers and the concentration of interferon-gamma as an indicator of cellular response in 28 selected from the research group. The studies included patients vaccinated only with BNT162b2 or convalescents (symptomatic and asymptomatic). Analyzes were conducted at weekly time points, i.e. at week 2 and week 3 after the first dose and 1-5 weeks after the second dose of the vaccine. Additionally, the paper presents regular monitoring of the antibody level of 8 laboratory workers vaccinated with BNT162b2 for 4 months.

The work of Szczepanek et al. 2022 concerned the determination of IgG anti-SARS-CoV-2 in a group of 954 employees of the Nicolaus Copernicus University in Toruń, of whom 511 underwent serological tests three times (beginning of May, end of June and end of September). Analyzes were conducted at these three time points using the four available COVID-19 vaccines. The anti-SARS-CoV-2 IgG level was also analyzed in terms of correlation with the age of the individuals (in the age groups <30 years, 30-40, 40-50, 50-65, 65+), sex and blood group. Anti-SARS-CoV-2 IgG level determinations were carried out until September 2021.

The paper reported for review presents the results of anti-SARS-CoV-2 IgG levels in 18,610 healthcare workers carried out for 1-12 months after full vaccination. It is important to emphasize the novelty and value of the analysis of the long-term vaccine effect and of a very large research group. The effects of four COVID-19 vaccines were also investigated here, although analyzes were conducted at different time points. It should also be mentioned that the analyzes carried out from September to December allowed us to obtain new and very important information. During this time period, we were able to test people vaccinated with the 3rd dose of the COVID-19 vaccine. What's more, it was possible to highlight a large group of people who had passed COVID-19 and did not know about it. This information was the basis for the estimated duration of vaccine protection.

Summing up, the conclusions from the previous studies concern the dynamics of changes in anti-SARS-CoV-2 antibody levels in the first months after vaccination, comparison of IgG titers in vaccinated and convalescents, comparison of mRNA and vector vaccine responses, and the advisability of introducing a third booster dose. This work focuses mainly on the comparison of the long-term anti-SARS-CoV-2 IgG status caused by the four vaccines available in Poland, the analysis of antibody levels after the intake of the third dose of the vaccine, and the estimation of the post-vaccination protection period against SARS-CoV-2 infection.

As you can see, it is not possible to mention and list the differences very briefly. Therefore, to prevent this description from being dominant in the presented work, we decided not to include it. Hope the reviewer will agree with us. We are sure that after reading each of the works, no one will have doubts about the individuality of each of them.

Finally, I recommend the authors double-check the full text for grammatical and typing errors.

Thank you very much for your valuable advice. The text has been reviewed and corrected again. Subsequently, it was subjected to professional linguistic proofreading at MDPI English Editing Services (see – ms - tracking changes).

Round 2

Reviewer 2 Report

Dear Authors,

I appreciate the effort you put into improving your article and the detailed answers which you have provided.

It is now more effectively structured and well-balanced, and I am definitely recommending its publication in light of its indisputable research value.

All the best with your future endeavors.

Sincerely,